# Kaempferide Enhances Chemosensitivity of Human Lung Adenocarcinoma A549 Cells Mediated by the Decrease in Phosphorylation of Akt and Claudin-2 Expression

**DOI:** 10.3390/nu12041190

**Published:** 2020-04-23

**Authors:** Hiroaki Eguchi, Toshiyuki Matsunaga, Satoshi Endo, Kenji Ichihara, Akira Ikari

**Affiliations:** 1Laboratory of Biochemistry, Department of Biopharmaceutical Sciences, Gifu Pharmaceutical University, Gifu 501-1196, Japan; 146008@gifu-pu.ac.jp (H.E.); sendo@gifu-pu.ac.jp (S.E.); 2Education Center of Green Pharmaceutical Sciences, Gifu Pharmaceutical University, Gifu 502-8585, Japan; matsunagat@gifu-pu.ac.jp; 3Nagaragawa Research Center, API Co., Ltd., Gifu 502-0071, Japan; ichihara-kenji@api3838.co.jp

**Keywords:** claudin, lung adenocarcinoma, chemoresistance, flavonoid

## Abstract

Claudins (CLDNs) play crucial roles in the formation of tight junctions. We have reported that abnormal expression of CLDN2 confers chemoresistance in the spheroids of human lung adenocarcinoma A549 cells. A food composition, which can reduce CLDN2 expression, may function to prevent the malignant progression. Here, we found that ethanol extract of Brazilian green propolis (EBGP) and kaempferide, a major component of EBGP, decrease CLDN2 expression. In the two-dimensional culture model, EBGP decreased the tight junctional localization of CLDN2 without affecting that of zonula occludens-1, an adaptor protein, and enhanced paracellular permeability to doxorubicin, a cytotoxic anticancer drug. EBGP reduced hypoxic stress, and enhanced the accumulation and sensitivity of doxorubicin in the spheroid of A549 cells. Kaempferide dose-dependently decreased CLDN2 expression, although dihydrokaempferide and pinocembrin did not. The phosphorylation of Akt, a regulatory factor of CLDN2 expression, was inhibited by kaempferide but not by dihydrokaempferide. The 2,3-double bond in the C ring may be important to inhibit Akt. Kaempferide decreased the mRNA level and promoter activity of CLDN2, indicating that it inhibits the transcription of CLDN2. In accordance with EBGP, kaempferide decreased the tight junctional localization of CLDN2 and increased a paracellular permeability to doxorubicin, suggesting that it diminished the paracellular barrier to small molecules. In addition, kaempferide reduced hypoxic stress, and enhanced the accumulation and sensitivity of doxorubicin in the spheroids. In contrast, dihydrokaempferide did not improve the sensitivity to doxorubicin. Further study is needed using an animal model, but we suggest that natural foods abundantly containing kaempferide are candidates for the prevention of the chemoresistance of lung adenocarcinoma.

## 1. Introduction

Lung cancer is one of the most lethal malignant tumors, and non-small cell lung cancer (NSCLC) accounts for approximately 85% of all lung cancer patients [1]. NSCLC is divided into three types: Adenocarcinoma, squamous cell carcinoma, and large cell carcinoma. A decline in smoking habits leads to a suppression of the incidence of squamous cell carcinoma, but the frequency of adenocarcinoma has increased. Adenocarcinoma is now the most common pathological type of NSCLC [2]. Targeted therapies with tyrosine kinase inhibitors against the epidermal growth factor receptor (EGFR-TKIs) and immune checkpoint inhibitors (ICIs) are effective to the most common treatment for NSCLC [3,4]. However, most patients treated with EGFR-TKI and ICIs show a resistance to chemotherapy around one year after the treatments. The formation of a tumor microenvironment consisting of fibroblasts, extracellular matrix, immune cells, and other cellular interactions in the body plays a key role in the development of chemoresistance [5], but the molecular mechanisms have not been understood well.

Tight junctions (TJs), which are the most apical constituent of lateral membrane in the epithelial cells, regulate paracellular transport of solutes and ions, and contribute to the cellular polarity, differentiation, proliferation, and migration [6,7]. Claudins (CLDNs) are essential components of TJs and comprise a large family of over 20 subtypes in mammals [8]. Abnormal expression of CLDNs has been reported in various solid tumor tissues, which is correlated with cancer progression and patient survival [9]. We recently reported that the expression of CLDN2 is increased in human lung adenocarcinoma tissues and adenocarcinoma-derived cultured cells, including A549 [10]. Proliferation and migration of A549 cells were decreased by CLDN2 knockdown [11,12]. Similarly, the elevation of CLDN2 expression is reported in other cancer tissues, such as the liver [13], colon [14], and stomach [15]. Therefore, CLDN2 may be one of the therapeutic targets for anticancer therapy, but there are few compounds that can reduce CLDN2 expression in lung adenocarcinoma cells.

Brazilian green propolis, which derives from the native plant *Baccharis dracunculifolia*, is one of the most commercialized propolis. The ethanol extract of Brazilian green propolis (EBGP) has a powerful anticancer activity [16]. The principal flavonoids of EBGP are kaempferide, kaempferol, dihydrokaempferide, pinocembrin, chrysin, and things like that [17]. In addition, artepillin C, baccharin, and drupanin, which are categorized as cinnamic acid derivatives, are also abundantly contained in EBGP [18]. However, the ingredients of EBGP with anticancer activities and the anticancer mechanisms have not been fully elucidated. We have reported that the transcriptional activity of CLDN2 is upregulated by the phosphatidylinositol 3-kinase (PI3K)/Akt/nuclear factor-κB [19], mitogen-activated protein kinase kinase (MEK)/extracellular signal-regulated kinase (ERK) [10], and signal transducer and activator of transcription 3 (Stat3) pathways [20]. These intracellular signaling pathways may be useful to search for compounds that can reduce CLDN2 expression in lung adenocarcinoma cells.

In the present study, we examined the effects of EBGP and its components on CLDN2 expression by Western blotting and real-time polymerase chain reaction (PCR) experiments in A549 cells. Cell localization and transcriptional activity of CLDN2 were investigated by immunofluorescence and promoter assays, respectively. Chemosensitivity was determined by the ATP assay using a three-dimensional (3D) culture model. Our data indicate that EBGP, which abundantly contains kaempferide, may be a candidate for preventing the chemoresistance of lung adenocarcinoma.

## 2. Experimental Section

### 2.1. Materials

Kaempferide, kaempferol, dihydrokaempferol, and pinocembrin were purchased from INDOFINE Chemical Company (Somerville, NJ, USA), Tokyo Kasei Kogyo (Tokyo Japan), Chemodex (Gallen, Switzerland), and Extrasynthese (Genay Cedex, France), respectively. Dihydrokaempferide and EBGP were provided from Nagaragawa Research Center (API Co., Ltd., Gifu, Japan). Flavonoids were dissolved in dimethyl sulfoxide. Goat anti-β-actin antibody and doxorubicin were purchased from Santa Cruz Biotechnology (Santa Cruz, CA, USA) and Wako Pure Chemical Industries (Osaka, Japan), respectively. Rabbit anti-CLDN1, rabbit anti-CLDN2, and mouse anti-zonula occludens-1 (ZO-1) antibodies were from Thermo Fisher Scientific (Rockford, IL, USA). All other reagents were of the highest purity available.

### 2.2. Cell Culture

Human lung adenocarcinoma-derived A549 cells were obtained from the RIKEN BRC through the National Bio-Resource Project of the MEXT (Ibaraki, Japan). The cells were cultured as described previously [21]. Cell viability in a two-dimensional (2D) culture model was measured using 4-[3-[4-iodophenyl]-2-4(4-nitrophenyl)-2*H*-5-tetrazolio-1,3-benzene disulfonate] (WST-1). Dimethysulfoxide (<0.5%) or ethanol (<1%) was added to control samples to exclude a solvent or carrier effect on cell responses. We confirmed that the solvent did not affect cell viability, the expression of CLDN, and barrier function in advance.

### 2.3. Western Blotting

Cell lysates were prepared as described previously [21]. Aliquots of cell lysates (30–60 μg) were applied to sodium dodecyl sulfate-polyacrylamide gel electrophoresis, and then blotted onto a poly (vinylidene fluoride) membrane using SemiDry transfer blot (Bio-Rad, Hercules, CA, USA). After blocking with 2% skim milk for 30 min, the membrane was in cubated with each primary antibody (1:1000 dilution) at 4 °C for 16 h, followed by a peroxidase-conjugated secondary antibody (1:3000 dilution) at room temperature for 1 h. Finally, the blots were incubated by ImmunoStar LD (Wako) or EzWestLumi plus (Atto Corporation, Tokyo, Japan), and scanned using a C-DiGit Blot Scanner (LI-COR Biotechnology, Lincoln, NE, USA). Band density was quantified using ImageJ software (National Institute of Health, Bethesda, MD, USA). The signals were normalized to the loading control (β-actin).

### 2.4. RNA Isolation and Quantitative Reverse Transcription-PCR

Total RNA was isolated from the cells using TRI reagent (Molecular Research Center, Cincinnati, OH, USA). Reverse transcription and quantitative real-time PCR were performed using specific primers against human CLDN2 and the threshold cycle for each PCR product was calculated as described previously [20].

### 2.5. Immunocytochemistry

The cellular localization of CLDN1, CLDN2, and ZO-1 was investigated by immunocytochemistry. The cells were incubated with each first antibody for 16 h at 4 °C, and then incubated with Alexa Fluor 488- and 555-conjugated antibodies for 1.5 h at room temperature. The florescence images were collected using an LSM 700 confocal microscope (Carl Zeiss, Jena, Germany).

### 2.6. Paracellular Barrier Function

Cells were cultured on transwell plates (0.4 μm pore size, 12 mm diameter) with polyester membrane inserts (Corning Incorporated, Corning, NY, USA). The barrier function of TJ was estimated by transepithelial electrical resistance (TER) and flux of doxorubicin [22,23].

### 2.7. Spheroid Model

Spheroids were formed in PrimeSurface96U multi-well plates (Sumitomo Bakelite, Tokyo, Japan). After 4 days, the circumference length of spheroids was measured using ImageJ software. The hypoxia conditions were measured using a hypoxia probe solution (LOX-1, Medical & Biological Laboratories, Nagoya, Japan) for which fluorescent signal is only activated by hypoxic conditions. The spheroids were incubated with LOX-1 or doxorubicin for 1 h at 4 °C. The fluorescence intensities of LOX-1 and doxorubicin in the spheroids were measured using a BZ-X800 fluorescence microscope (Keyence, Osaka, Japan). The viability of spheroid cells was examined using a CellTiter-Glo 3D Cell Viability Assay kit (Promega, Madison, WI, USA). The intensity of chemiluminescence was measured using a Luminescencer Octa AB-2270 (Atto, Tokyo, Japan).

### 2.8. Statistical Analysis

Data are presented as means ± S.E.M. Comparisons between two groups were made using Student’s *t* test. Differences between groups were analyzed by one-way or two-way analysis of variance, and corrections for multiple comparison were made using Tukey’s multiple comparison test. Statistical analyses were performed using KaleidaGraph version 4.5.1 software (Synergy Software, PA, USA). Significant differences were assumed at *p* < 0.05.

## 3. Results

### 3.1. Effect of EBGP on CLDN2 Expression in A549 Cells

EBGP shows an anticancer effect in rats [24], but the mechanism has not been fully understood. We reported that CLDN2 is involved in the malignant A549 cells [11,12]. The protein level of CLDN2 was decreased by EBGP in a dose-dependent manner (Figure 1A,B). EBGP did not show cytotoxicity until a concentration of 50 μg/mL under our experimental conditions (Figure 1C). These results indicate that the decrease in CLDN2 expression by EBGP may not be related to cytotoxicity. The mRNA level of CLDN2 was also decreased by EBGP in a dose-dependent manner (Figure 1D). EBGP may decrease CLDN2 expression in A549 cells mediated by the inhibition of the transcriptional activity of CLDN2.

### 3.2. Effect of EBGP on the Cell Localization of CLDN2 and Transepithelial Permeability

Immunofluorescence measurements indicated that CLDN2 is colocalized with zonula occludens-1 (ZO-1) at the cell–cell border area (Figure 2A). EBGP decreased the red signal of CLDN2 without affecting the localization of ZO-1. CLDN2 forms a paracellular cation channel permeable to Na^+^, and the CLDN2-expressing cells show lower transepithelial electrical resistance (TER) [25,26]. We estimated the function of the TJ barrier by measuring TER and the transepithelial flux of doxorubicin. EBGP significantly increased TER, whereas EBGP increased the transepithelial fluxes of doxorubicin (Figure 2B,C), suggesting that CLDN2 may function as a cation channel and barrier to small molecules. These results are consistent with those in the CLDN2 knockdown experiments [27].

### 3.3. Increase in Doxorubicin-Induced Cytotoxicity by EBGP in a Spheroid Model.

Cancer cells form a microenvironment, which facilitates the chemoresistance. The 3-D spheroid model is useful to study chemoresistance. To clarify the effect of EBGP on chemosensitivity in A549 spheroid cells, we investigated the size, hypoxic level, and cell viability. Spheroid size was unchanged by EBGP, but the hypoxic level inside the spheroids was significantly decreased (Figure 3A). In contrast, the ATP content, which correlates with the cell viability, was increased by EBGP. The fluorescence intensity of doxorubicin in the spheroids increased in a dose-dependent manner, which was further enhanced by EBGP (Figure 3B). In addition, EGFP enhanced the doxorubicin-induced toxicity (Figure 3C). These results indicate that EBGP may attenuate hypoxic stress inside the spheroids, which contributes to chemoresistance [28].

### 3.4. Effects of Kaempferol and Kaempferide on CLDN2 Expression in A549 Cells

A variety of flavonoids, especially kaempferol and kaempferide, are abundantly contained in EBGP [17]. Flavonoids have a common structure of two aromatic rings connected to three carbons. Kaempferide differs with kaempferol in only one substituent of the 4′-hydroxy group in ring B. (Figure 4A). Neither kaempferol nor kaempferide decreased cell viability, but kaempferide slightly increased the viability at 50 µM (Figure 4B). Kaempferide showed a potent inhibitory effect on CLDN2 expression and the effect was significant above 1 µM (Figure 4C). In contrast, the protein level of CLDN2 was decreased by kaempferol, but the effect was significant above 10 µM. Therefore, we investigated the structure–activity relationships of flavonoids and the regulatory mechanism of CLDN2 expression by kaempferide.

### 3.5. Structure–Activity Relationship of Flavonoids in Decreasing CLDN2 Expression

To clarify the involvement of the 2,3-double bond in the C-ring of flavonoids, we investigated the effects of dihydrokaempferol, dihydrokaempferide, and pinocembrin on CLDN2 expression (Figure 5). These flavonoids are also contained in EBGP [17]. Dihydrokaempferide slightly decreased the cell viability at a concentration of 50 μM, whereas neither dihydrokaempferol nor pinocembrin did. The CLDN2 expression was not decreased by dihydrokaempferol, dihydrokaempferide, and pinocembrin below 50 µM. These results indicated that the 2,3-double bond in the C-ring may be required for the inhibition of CLDN2 expression.

### 3.6. Decrease in the Phosphorylation Level of Akt by Kaempferide

The CLDN2 expression is upregulated by several signaling pathways, including the PI3K/Akt pathway. The phosphorylation level of Akt was decreased by kaempferide, whereas that of phosphoinositide-dependent kinase 1 (PDK1), a kinase upregulating the phosphorylation of Akt, was not (Figure 6). In contrast, neither p-PDK1 nor p-Akt were significantly inhibited by dihydrokaempferide. EBGP decreased the p-Akt level without affecting the p-PDK1 level, which are similar to those in the kaempferide-treated cells. These results indicate that the decrease in the phosphorylation of Akt may be involved in the reduction of CLDN2 expression by EBGP and kaempferide.

### 3.7. Effect of Kaempferide on the Transcription Activity of CLDN2

The mRNA level of CLDN2 was decreased by kaempferide but not by dihydrokaempferide (Figure 7A). Both EBGP and kaempferide significantly decreased the promoter activity of CLDN2 (Figure 7B). These results coincide with those of the Western blotting. Therefore, kaempferide may reduce the protein level of CLDN2 mediated by inhibition of the PI3K/Akt pathway and the transcriptional activity of CLND2.

### 3.8. Effect of EBGP on the Cell Localization of CLDN2 and Transepithelial Permeability

Immunofluorescence measurement showed that kaempferide decreases the red signal of CLDN2 without affecting the localization of ZO-1 as well as EBGP (Figure 8A). Kaempferide increased TER (Figure 8B), indicating that the paracellular cation transport may be reduced by the decrease in CLDN2 expression. In contrast, kaempferide increased the transepithelial flux of doxorubicin (Figure 8C), indicating that the barrier function against a small molecule may be reduced by the decrease in CLDN2 expression. These results are consistent with those of EBGP.

### 3.9. Increase in Doxorubicin-Induced Cytotoxicity by Kaempferide in a Spheroid Model

To clarify whether kaempferide and kaempferol affect the chemosensitivity, we examined the effects of these compounds on the size, hypoxic level, and accumulation and toxicity of doxorubicin using a 3-D spheroid model. Spheroid size was changed by neither kaempferide nor dihydrokaempferide, but thee hypoxic level was significantly decreased by kaempferide. In contrast, dihydrokaempferide changed neither the spheroid size nor the hypoxic level. The ATP content was only increased by kaempferide (Figure 9A), indicating that the reduction of the hypoxic level may increase the cell viability. The fluoresce intensity and toxicity of doxorubicin were enhanced by kaempferide (Figure 9B,C). These results consist with those of EBGP, indicating that kaempferide may be an active ingredient of EBGP. To support the involvement of CLDN2 expression in chemoresistance, we examined the effect of CLDN2 overexpression on the accumulation and toxicity of doxorubicin. Kaempferide enhanced the accumulation and toxicity of doxorubicin, which were inhibited by CLDN2 overexpression (Figure 10).

## 4. Discussion

We recently reported that CLDN2 may be involved in the development of lung adenocarcinoma [11,12]. The elevation of CLDN2 expression has been reported in various solid cancers [13,14,15]. EBGP has recently been reported to possess potent anticancer activities against many tumor cells [16]. In addition, EBGP improves chemoresistance in prostate cancer cells [29,30]. However, the bioactive compounds may vary according to the season of collection, region, climate, and bee species [31,32]. To clarify the mechanism of propolis, it would be important to investigate the effect of active compounds earlier. We have reported that some components of propolis, such as kaempferol, quercetin, and caffeic acid phenethyl ester, have an ability to reduce CLDN2 expression in A549 cells [20,33,34]. However, the amounts of these active components in propolis are low. In the present study, we found that CLDN2 expression is decreased by kaempferide, which is highly contained in EBGP (1700 mg/100 g) [35] compared to other flavonoids.

A variety of flavonoids, including kaempferol and quercetin, have antioxidant and pro-oxidant activities, which are involved in the anticancer effects [36]. Is CLDN2 expression affected by an intracellular-reduced condition? Kaempferol has a potent antioxidant property (EC_50_ 14.6 μg) similar to trolox (EC_50_ 12.5 μg). In contrast, the antioxidant ability of kaempferide is low (EC_50_ 97 μg) [37]. Furthermore, we recently reported that *N*-acetyl-L-cysteine, a well-known antioxidant drug, did not decrease CLDN2 expression in A549 cells [22]. Other molecular mechanisms without an antioxidant effect may be involved in the kaempferide-induced decrease in CLDN2 expression.

Some flavonoids can enhance chemosensitivity in the cancer spheroid model, but the mechanism is unclarified in detail [38]. Therefore, it is important to clarify the molecular mechanism of flavonoids. The expression of CLDN2 is upregulated by several intracellular signaling pathways, including the PI3K/Akt pathway in A549 cells [10,20]. LY-294002, an inhibitor of PI3K, decreases mRNA levels and promoter activity of CLDN2 [19]. Kaempferide decreased the p-Akt level without affecting the total amount of Akt (Figure 6). Akt is a downstream effector of PI3K and PDK1, and PDK1 can phosphorylate Akt [39]. Kaempferide did not decrease the p-PDK1 level, indicating that it may directly inhibit the phosphorylation of Akt. Similar results were observed by EBGP. In contrast, dihydrokaempferide, which lacks a 2,3-double bond in the C ring of the flavonoid structure, did not decrease p-Akt and CLDN2 expression levels. The 2,3-double bond in the C ring may be necessary to decrease the phosphorylation of Akt and CLDN2 expression. There are other compounds in EBGP that could produce additional or synergistic effects on CLDN2 expression. Further study is needed to clarify the combination effects of compounds.

Hypoxic stress in spheroids is involved in the malignant transformation of cancer cells [28]. Hypoxic stress induces hypoxia inducible factor-1 (HIF-1) expression. Under normoxic conditions, HIF-1 is ubiquitinated and degraded by proteasome. HIF-1 contributes to tumor angiogenesis and metastasis mediated via various mechanisms, such as induction of the drug efflux pumps and drug-metabolizing enzymes, and DNA epigenetic states, leading to the development of drug resistance [40]. Both EBGP and kaempferide reduced the hypoxic condition and increased cell viability in the spheroid of A549 cells (Figure 3 and Figure 9). The elevation of viability may cause a reduction of hypoxic stress. The sensitivity to doxorubicin was also enhanced by EBGP and kaempferide. The oxygen concentration in spheroids depends on the balance between the amount of oxygen supply and the consumption by cells [41]. At present, we do not know how EBGP and kaempferide reduce the hypoxic level, but kaempferide may be useful to prevent the development of lung adenocarcinoma.

## 5. Conclusions

In the present study, we found that EBGP and kaempferide, a most abundant flavonoid in EBGP, decrease CLDN2 expression in A549 cells. Kaempferide reduced the hypoxic condition in spheroid cells and enhanced the accumulation and sensitivity of doxorubicin. EBGP and foods rich in kaempferide may be candidates for the prevention of malignant progression of lung adenocarcinoma. In a future study, we need to clarify the effects of EBGP and kaempferide using an in vivo animal model.

## Figures and Tables

**Figure 1 nutrients-12-01190-f001:**
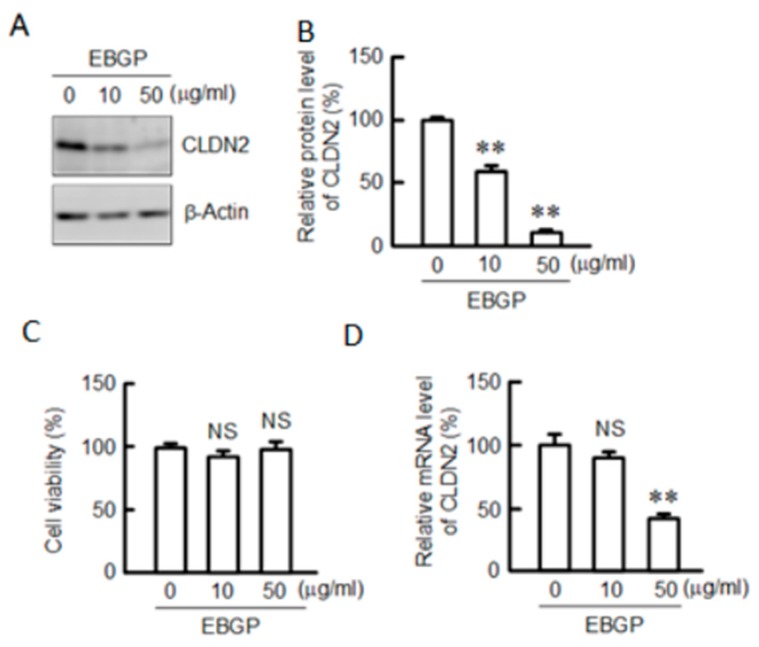
Effect of ethanol extract of Brazilian green propolis (EBGP) on the viability and expression of claudin-2 (CLDN2) in A549 cells. (**A**) Cells were incubated with 0, 10, and 50 μg EBGP for 24 h, followed by incubation with 4-[3-[4-iodophenyl]-2-4(4-nitrophenyl)-2*H*-5-tetrazolio-1,3-benzene disulfonate] (WST-1) reagent. Cell viability is represented as a percentage relative to 0 μg/mL. (**B**,**C**) The cell lysates were immunoblotted with anti-CLDN2 and anti-β-actin antibodies. The protein level of CLDN2 is represented as a percentage relative to 0 μg (**D**). The mRNA level of CLDN2 was determined by quantitative real-time PCR, and is represented as a percentage relative to 0 μg/mL. *n* = 3–4. ** *p* < 0.01 and NS *p* > 0.05 compared with 0 μg/mL (one-way analysis). Protein (F_3,8_ = 201.24, *p* < 0.0001), viability (F_6,9_ = 0.05, *p* = 0.953646), and mRNA (F_3,8_ = 28.31, *p* = 0.00023).

**Figure 2 nutrients-12-01190-f002:**
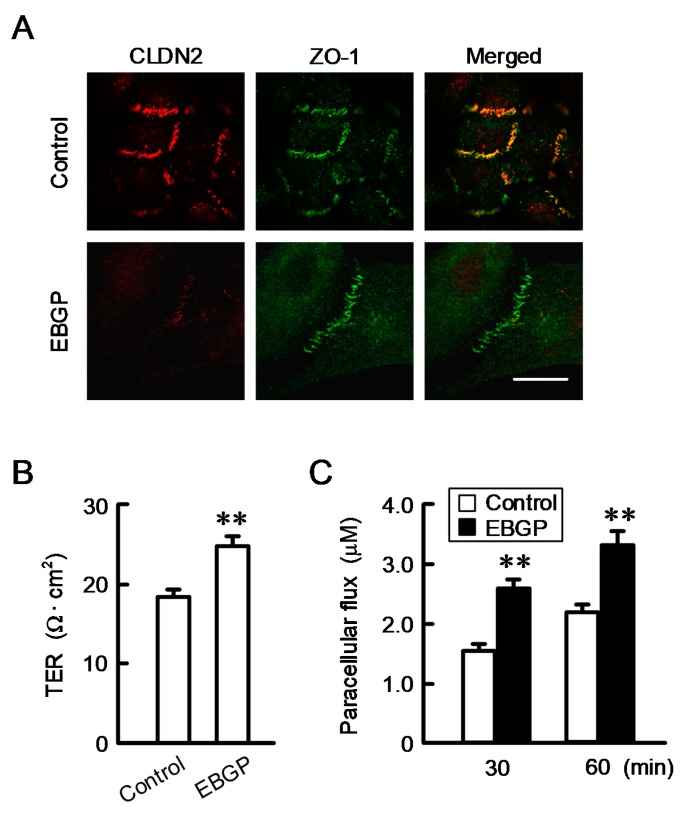
Effect of EBGP on cellular localization of CLDN2 and barrier function. (**A**) Cells cultured on cover glasses were incubated in the absence (control) and presence of 50 μg/mL EBGP for 24 h. The cells were stained with anti-CLDN2 (red) and anti-zonula occludens-1 (ZO-1) (green) antibodies. Images were taken using the confocal laser microscope equipped with × 100 objective lens. Merged images are shown on the right. Scale bar represents 10 μm. (**B**,**C**) Cells cultured on transwell inserts were incubated in the absence and presence of 50 μg/mL EBGP for 24 h. TER was measured using a volt ohmmeter. Doxorubicin (10 μM) was added to the apical compartment. After incubation at 4 °C for 60 min, the solution in the basal compartment was collected, followed by measurement of the fluorescence intensity using an Infinite F200 Pro microplate reader. *n* = 4. ** *p* < 0.01 compared with control (Student’s *t* test).

**Figure 3 nutrients-12-01190-f003:**
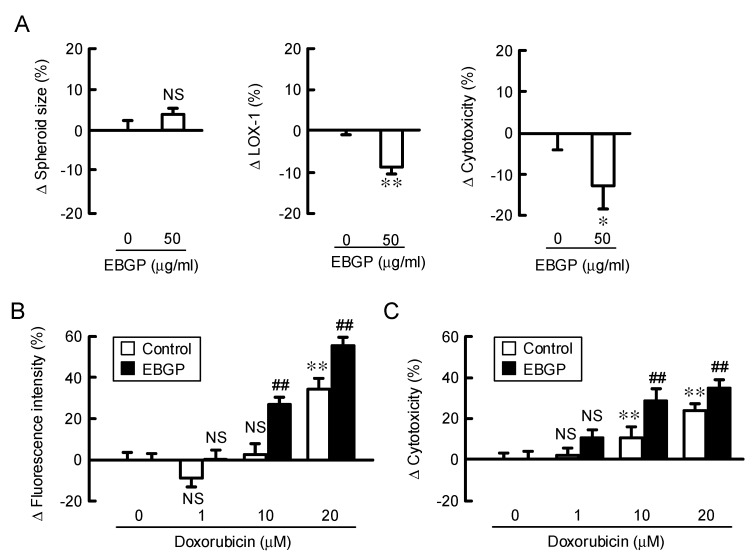
Effect of EBGP on the sensitivity to doxorubicin in the spheroids. (**A**) The spheroids were incubated in the absence and presence of 50 μg of EBGP for 24 h. Spheroid size, fluorescence intensity of a hypoxia probe solution (LOX-1), and cell viability are represented as a percentage relative to 0 μg/mL. ** *p* < 0.01, * *p* < 0.05, and NS *p* > 0.05 compared with 0 μg/mL EBGP (Student’s *t* test). (**B**) The spheroids were pre-incubated with 50 μg of EBGP for 24 h, followed by incubation with doxorubicin for 60 min. The fluorescence intensity of doxorubicin in the spheroids was examined using a BZ-X800 fluorescence microscopy. (**C**) After incubation of the spheroids with doxorubicin in the absence and presence of 50 μg/mL of EBGP for 24 h, cell viability was measured using a CellTiter-Glo 3D Cell Viability Assay kit. The fluorescence intensity of doxorubicin and cell viability are represented as a percentage relative to 0 μM doxorubicin. *n* = 3–6. ** *p* < 0.01 compared with 0 μg/mL EBGP. ^##^
*p* < 0.01 compared with control. NS *p* > 0.05 compared with 0 μg/mL EBGP or control (Two-way analysis). Fluorescence intensity (F_1,31_ = 145.43, *p* < 0.0001) and cytotoxicity (F_1,23_ = 57.45.45, *p* < 0.0001).

**Figure 4 nutrients-12-01190-f004:**
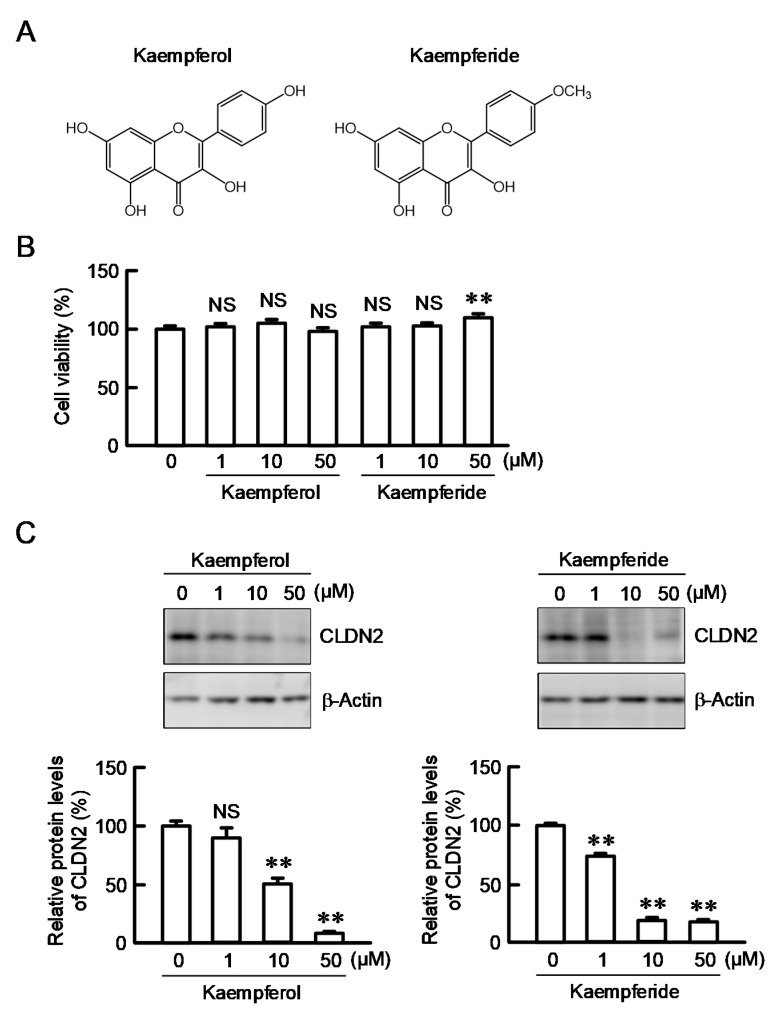
Decrease in CLDN2 expression by kaempferol and kaempferide. (**A**) Structures of kaempferol and kaempferide. (**B**,**C**) Cells were incubated with 0, 1, 10, and 50 μM kaempferol or kaempferide for 24 h. Cell viability was assessed by the WST-1 assay and represented as a percentage relative to 0 μM. The protein level of CLDN2 was examined by Western blotting and represented as a percentage relative to 0 μM. β-Actin was used as an internal control. *n* = 3–8. ** *p* < 0.01 and NS *p* > 0.05 compared with 0 μM (one-way analysis). Viability (F_6,22_ = 5.19, *p* = 0.00184), protein in kaempferol (F_3,8_ = 52.71, *p* < 0.0001), and protein in kaempferide (F_3,8_ = 490.69, *p* < 0.0001).

**Figure 5 nutrients-12-01190-f005:**
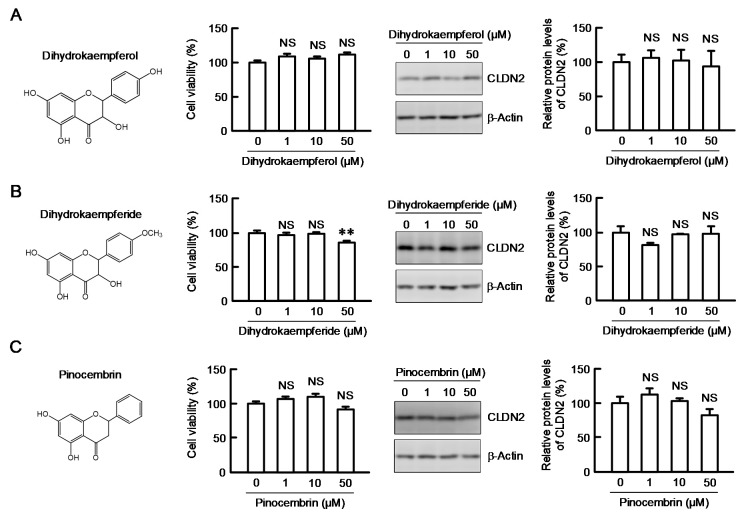
Effects of dihydrokaempferide derivatives on viability and CLDN2 expression. Cells were incubated with 0, 1, 10, and 50 μM dihydrokaempferol (**A**), dihydrokaempferide (**B**), or pinocembrin (**C**) for 24 h. The structures of these derivatives are shown in the left. Cell viability was assessed by WST-1 assay and represented as a percentage relative to 0 μM. The protein level of CLDN2 was examined by Western blotting and represented as a percentage relative to 0 μM. *n* = 3–8. ** *p* < 0.01 and NS *p* > 0.05 compared with 0 μM (One-way analysis). Viability in dihydrokaempferol (F_6,11_ = 4.72, *p* = 0.02357), viability in dihydrokaempferide (F_6,16_ = 17.65, *p* < 0.0001), viability in pinocembrin (F_6,11_ = 9.43, *p* = 0.00892), protein in dihydrokaempferol (F_3,11_ = 0.11, *p* = 0.95341), protein in dihydrokaempferide (F_3,11_ = 1.31, *p* = 0.3379), protein in pinocembrin (F_3,11_ = 2.34, *p* = 0.14872).

**Figure 6 nutrients-12-01190-f006:**
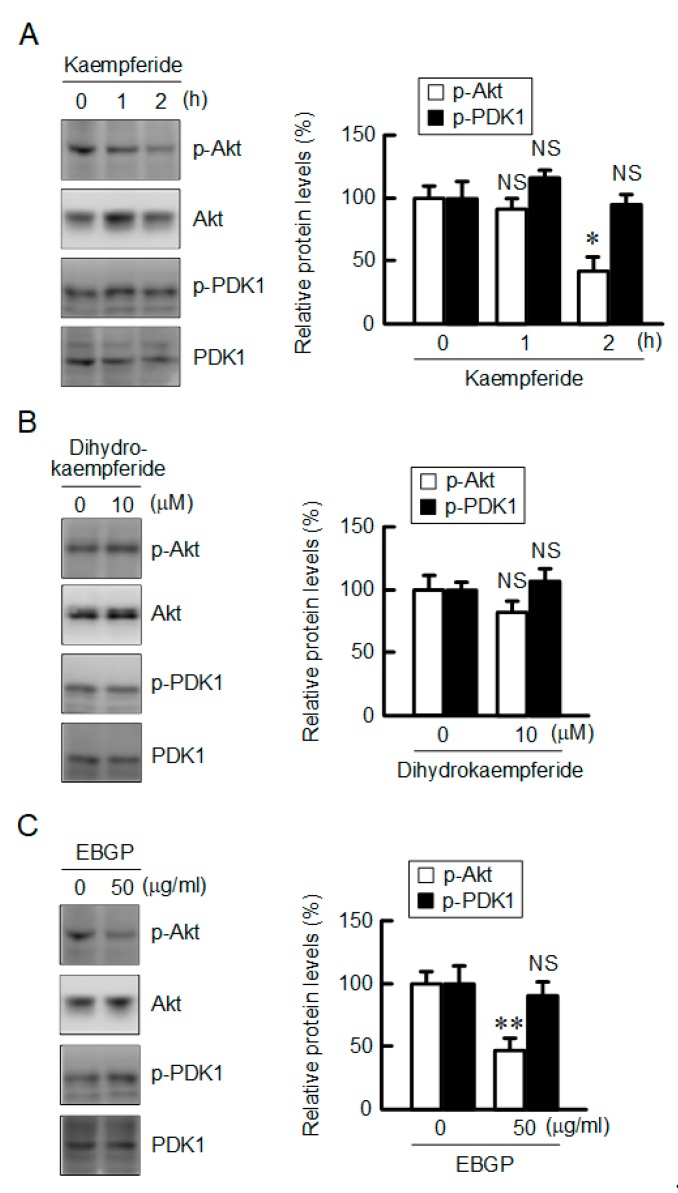
Inhibition of phosphorylation of Akt by EBGP and kaempferide. Cells were incubated with 10 μM kaempferide for 0, 1, and 2 h (**A**), or 10 μM dihydrokaempferide (**B**) and 50 μg/mL EBGP for 2 h (**C**). The protein levels of p-Akt, Akt, p-PDK1, and PDK1 were examined by Western blotting and represented as a percentage relative to 0 h, 0 μM, or 0 μg/mL. *n* = 3–4. ** *p* < 0.01, * *p* < 0.05, and NS *p* > 0.05 compared with 0 h (one-way analysis), 0 μM, or 0 μg/mL (Student’s *t* test). Time course (F_3,6_ = 10.47, *p* = 0.01106).

**Figure 7 nutrients-12-01190-f007:**
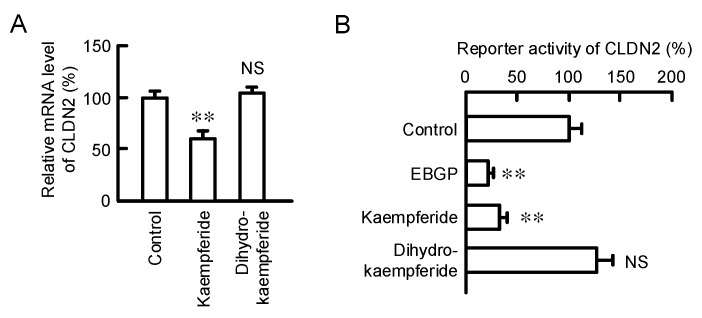
Effect of kaempferide on the transcriptional activity of CLDN2. (**A**) Cells were incubated with 10 μM kaempferide or 10 μM dihydrokaempferide for 6 h. The mRNA level of CLDN2 was examined by real-time PCR, and represented as a percentage relative to the control. (**B**) Cells were transfected with reporter vector of human CLDN2 and pRL-TK, an internal control. After treatment with 10 μg/mL EBGP, 10 μM kaempferide, or 10 μM dihydrokaempferide for 6 h, the reporter activity was measured by the Dual-Luciferase Reporter Assay System and represented as a percentage relative to control. *n* = 4–5. ** *p* < 0.01 and NS *p* > 0.05 compared with control (One-way analysis). mRNA (F_4,9_ = 18.16, *p* = 0.00069) and reporter activity (F_4,15_ = 24.45, *p* < 0.0001).

**Figure 8 nutrients-12-01190-f008:**
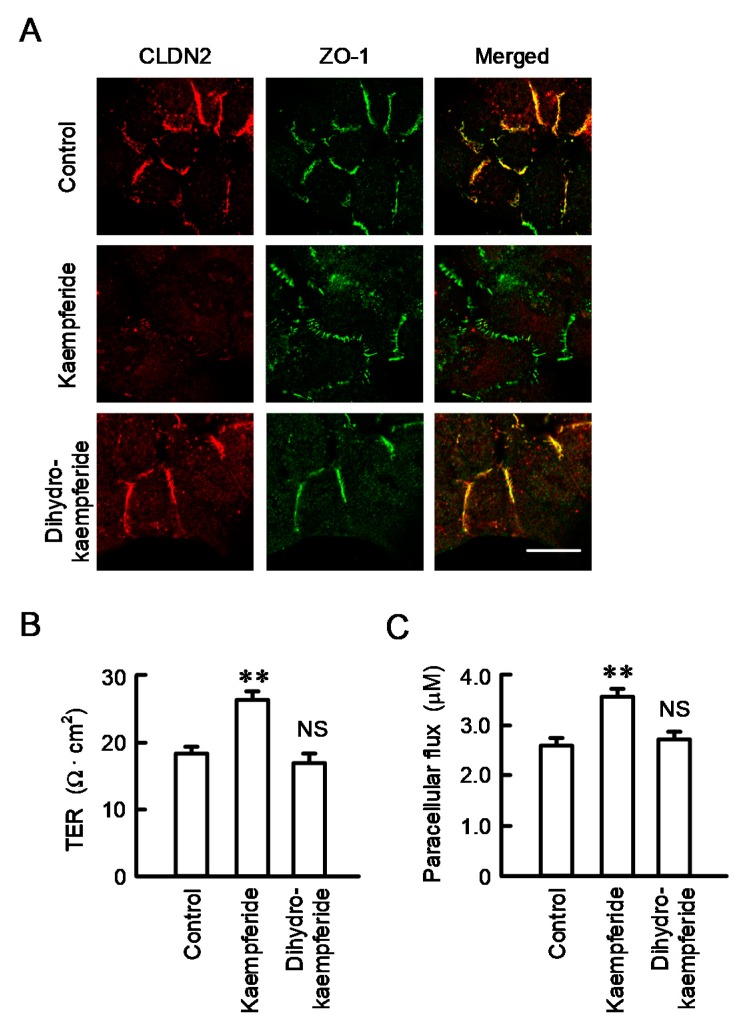
Effect of kaempferide on cellular localization of CLDN2 and barrier function. (**A**) Cells cultured on cover glasses were incubated in the absence and presence of 10 μM kaempferide or 10 μM dihydrokaempferide for 24 h. The cells were stained with anti-CLDN2 (red) and anti-ZO-1 (green) antibodies. Images were taken using the confocal laser microscope equipped with × 100 objective lens. Merged images are shown on the right. Scale bar represents 10 μm. (**B**,**C**) Cells cultured on transwell inserts were incubated in the absence and presence of 10 μM kaempferide or 10 μM dihydrokaempferide for 24 h. TER was measured using a volt ohmmeter. doxorubicin (10 μM) was added to the apical compartment. After incubation at 4 °C for 60 min, the solution in the basal compartment was collected, followed by measurement of fluorescence intensity. *n* = 4–8. ** *p* < 0.01 and NS *p* > 0.05 compared with control (One-way analysis). TER (F_4,9_ = 27.02, *p* = 0.0016) and paracellular flux (F_4,14_ = 13.90, *p* = 0.00047).

**Figure 9 nutrients-12-01190-f009:**
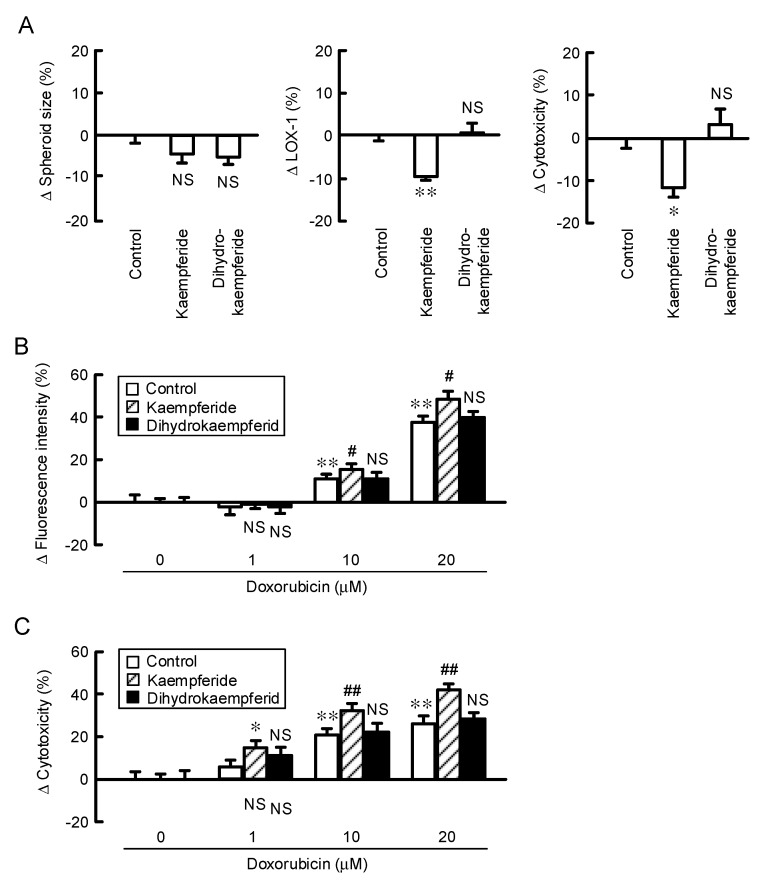
Increase in the sensitivity of spheroid cells to doxorubicin by kaempferide. (**A**) The spheroids were incubated in the absence (control) and presence of 10 μM kaempferide or 10 μM dihydrokaempferide for 24 h. Spheroid size, fluorescence intensity of LOX-1, and cell viability are represented as a percentage relative to the control. ** *p* < 0.01, * *p* < 0.05, and NS *p* > 0.05 compared with the control (one-way analysis). Spheroid size (F_2,16_ = 1.71, *p* = 0.21639), LOX-1 (F_2,11_ = 8.31, *p* = 0.00902), and cytotoxicity (F_2,10_ = 6.28, *p* = 0.02293). (**B**) The spheroids were pre-incubated with 10 μM kaempferide or 10 μM dihydrokaempferide for 24 h, followed by incubation with doxorubicin for 60 min. The fluorescence intensity of doxorubicin in the spheroids was examined using a BZ-X800 fluorescence microscopy. (**C**) After incubation of the spheroids with doxorubicin in the absence and presence of 10 μM kaempferide or 10 μM dihydrokaempferide for 24 h, cell viability was measured using a CellTiter-Glo 3D Cell Viability Assay kit. The fluorescence intensity of doxorubicin and cell viability are represented as a percentage relative to 0 μM doxorubicin. *n* = 4–6. ** *p* < 0.01, * *p* < 0.05, and NS *p* > 0.05 compared with 0 μM. ^##^
*p* < 0.01 and ^#^
*p* < 0.05 compared with control (Two-way analysis). Fluorescence intensity (F_2,59_ = 4.54, *p* = 0.01563) and cytotoxicity (F_2,35_ = 9.27, *p* = 0.00212).

**Figure 10 nutrients-12-01190-f010:**
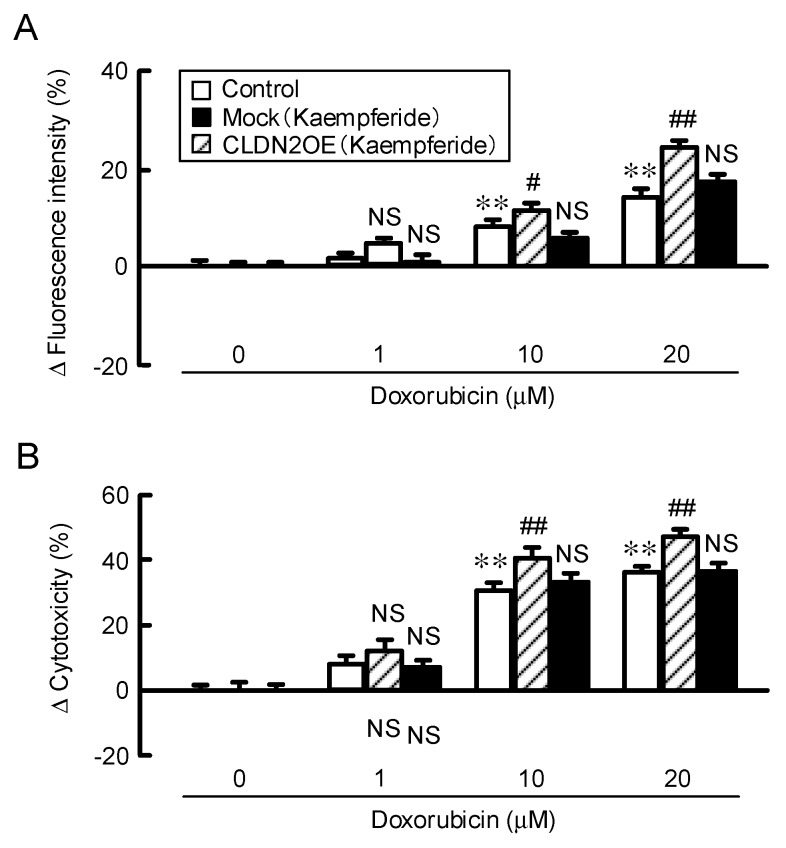
Elevation of chemoresistance by overexpression of CLDN2. (**A**) The spheroid cells were transfected with mock or CLDN2 expression vector followed by incubation with 10 μM kaempferide for 24 h. (**A**) The cells were incubation with doxorubicin for 60 min. The fluorescence intensity of doxorubicin in the spheroids was examined using a BZ-X800 fluorescence microscopy. (**B**) Cell viability was measured using a CellTiter-Glo 3D Cell Viability Assay kit. The fluorescence intensity of doxorubicin and cell viability are represented as a percentage relative to 0 μM doxorubicin. *n* = 5–8. ** *p* < 0.01 compared with 0 μM doxorubicin. ^##^
*p* < 0.01, ^#^
*p* < 0.05, and *NS p* > 0.05 compared with control (Two-way analysis). Fluorescence intensity (F_2,59_ = 22.07, *p* < 0.0001) and cytotoxicity (F_2,47_ = 42.71, *p* < 0.0001).

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
