# Peer review of "Kaempferide Enhances Chemosensitivity of Human Lung Adenocarcinoma A549 Cells Mediated by the Decrease in Phosphorylation of Akt and Claudin-2 Expression"

_nutrients, 2020, doi:10.3390/nu12041190_

Round 1

Reviewer 1 Report

In this study, Eguchi et al.  found that ethanol extract of Brazilian  Green propolis (EBGP) and kaempferide (K-ride), a major component of EBGP, decrease CLDN2 expression in lung A549 cells.  The authors previously made a similar observation in HaCaT Cells.  They attribute this effect to the inhibition of phosphorylation of Akt, although they have only shown a correlation in this study.  Also, they did not examine EBGP, only Kride.   Specific changes for improving the manuscript are listed  below:

Abstract, final sentence:  This global conclusion is not supported by the data from this study.  The authors have not quantified “malignant progression of lung adenocarcinoma.”

Introduction, page 2, line 76:  The final statement of the Introduction is not appropriate and should  be deleted.  The authors have not measured “adjuvant therapy”.

Results, page 4, line 136:  Rewrite—this currently does not make sense.

Results, page 4, line 154 and Fig 2A:  The authors state that EDGP did not affect localization  of  ZO-1, but it appears quite different in the treated cells.

Page 5, line 170:  The authors do not explain why they switched to the spheroid model halfway through their studies—or why  they did not use it from the beginning.  Was this the same cell type (A549)?

Page 5,lines 170-6 and Fig. 3:  The abbreviation LOX is not defined.  The LOX assay in the Methods is not satisfactorily explained.  Fig 3B and 3C:  These should be analyzed by 2-way ANOVA, so that the effects of both DXR and EBGP can be determine, and if there are any significant interactions.

Page 6, lines 189-90:  this is not clear

Page 6, line 190-1:  This statement is not supported by the figure.

Page 8, line 220:  Does EBGP also inhibit Akt phosphorylation?

Page 11, line 266-71, and Fig 9:  Again, the authors do not explain why they switched to the spheroid model halfway through their studies—or why  they did not use it from the beginning.  Was this the same cell type (A549)?

Page 12, line 299:  What is the difference between EEP and EBGP?

Page 13, line 343:  This statement is not supported by the data.

The manuscript needs to be edited for proper English.  Some deficiencies are listed above, but there are more.

Author Response

We thank you very much for your careful reading of our manuscript and valuable comments.

Comment 1

Abstract, final sentence:  This global conclusion is not supported by the data from this study. The authors have not quantified “malignant progression of lung adenocarcinoma.”

Answer

  Following your suggestion, we modified the last sentence. Please see line 30.

Comment 2

Introduction, page 2, line 76:  The final statement of the Introduction is not appropriate and should be deleted.  The authors have not measured “adjuvant therapy”.

Answer

  Following your suggestion, we modified the last sentence. Please see line 75.

Comment 3

Results, page 4, line 136:  Rewrite—this currently does not make sense.

Answer

  Following your suggestion, we modified the paragraph. Please see line 138.

Comment 4

Results, page 4, line 154 and Fig 2A:  The authors state that EBGP did not affect localization of ZO-1, but it appears quite different in the treated cells.

Answer

  Following your suggestion, we replaced the representative images. Please see new figure 2.

Comment 5

Page 5, line 170:  The authors do not explain why they switched to the spheroid model halfway through their studies—or why they did not use it from the beginning. Was this the same cell type (A549)?

Answer

  Following your point out, we described the reason why we switched to the spheroid. The 2D model is useful to clarify the expression level of protein and paracellular flux of small molecules. Therefore, we used 2D model at the beginning. In contrast, the 3D spheroid model is useful to study chemoresistance. Please see line 174. A549 cells were used in the studies of spheroid model.

Comment 6

Page 5, lines 170-6 and Fig. 3: The abbreviation LOX is not defined. The LOX assay in the Methods is not satisfactorily explained. Fig 3B and 3C: These should be analyzed by 2-way ANOVA, so that the effects of both DXR and EBGP can be determine, and if there are any significant interactions.

Answer

  LOX-1 is a product name of fluorescence probe and is not abbreviation. We described the method of LOX-1 assay in detail. Please see line 122. Following your suggestion, the effects of both DXR and EBGP were analyzed by 2-way ANOVA.

Comment 7

Page 6, lines 189-90:  this is not clear

Answer

  Following your suggestion, we modified the sentence. Please see line 197.

Comment 8

Page 6, line 190-1:  This statement is not supported by the figure.

Answer

  Following your suggestion, we corrected the sentence. Please see line 199.

Comment 9

Page 8, line 220:  Does EBGP also inhibit Akt phosphorylation?

Answer

  Following your point out, we performed additional experiments and found that EBGP inhibits Akt phosphorylation. Please see new figure 6.

Comment 10

Page 11, line 266-71, and Fig 9:  Again, the authors do not explain why they switched to the spheroid model halfway through their studies—or why they did not use it from the beginning.  Was this the same cell type (A549)?

Answer

  Following your point out, we described the reason. Please see line 279.

Comment 11

Page 12, line 299:  What is the difference between EEP and EBGP?

Answer

  Thank you very much for your point out. We corrected it.

Comment 12

Page 13, line 343:  This statement is not supported by the data.

Answer

  Thank you very much for your point out. We modified the sentence.

Reviewer 2 Report

This paper reports that propolis extract and kemperide increased sensitivity of adenocarcinoma A549 cells to doxorubicin (DXR), a cytotoxic anticancer drug, by decreaing claudin-2 expression. The study is well done and the conclusion is reasonably supprted by the experimental results. 

Minor commnts, 

  1. Please use chemical name as is. eg. kempferol, kemperide, doxorubicin and etc. No abbreviation is needed for single-word compound names.
  2. The effects of prolpois extract on the cell viability in the absence of doxorubicin are contradictory in Fig 3 A and C.
  3. Please show the efefct of kemperide on the mRNA level of claudin-2.
  4. Please show data for claudin-2 protein and mRNA levels in Fig 10.
  5. In Fig. 6. Please how time course of PI3k, PDK1 and Akt phosphorylation including earlier time points.
  6. It is also needed to examine whether Akt inhibitors can abrogate the effects kemperide.
  7. It is needed to explain how propolis extract and kemperide can decrease claudin-2 protein and mRNA levels. 

Overall the paper is interesting althouth the potency of kemperide seems marginal and in vivo test is additionally needed. The limitation of this study should be discussed carefully.  

Author Response

We thank you very much for your careful reading of our manuscript and valuable comments.

Minor comments

Comment 1

Please use chemical name as is. eg. kaempferol, kaempferide, doxorubicin and etc. No abbreviation is needed for single-word compound names.

Answer

Following your suggestion, we rewrote chemical name of kaempferol, kaempferide, doxorubicin, and so on.

Comment 2

The effects of propolis extract on the cell viability in the absence of doxorubicin are contradictory in Fig 3 A and C.

Answer

Following your suggestion, we corrected it. The cell viabilities in control and EBGP-treated cells are represented as a percentage relative to 0 mM doxorubicin, respectively, in figure 3C.

Comment 3

Please show the effect of kaempferide on the mRNA level of claudin-2.

Answer

  The effect of kaempferide on the mRNA level of claudin-2 is shown in figure 7.

Comment 4

Please show data for claudin-2 protein and mRNA levels in Fig 10

Answer

  Following your suggestion, we performed additional experiments. Unfortunately, we could not detect the expression of cladin-2 under the present experimental conditions.

Comment 5

In Fig. 6. Please how time course of PI3k, PDK1 and Akt phosphorylation including earlier time points.

Answer

  Following your suggestion, we performed additional experiments. The phosphorylation of Akt was not significantly decreased by kaempferide at 1 h, but decreased at 2 h. Please see new figure 6.

Comment 6

It is also needed to examine whether Akt inhibitors can abrogate the effects kaempferide.

Answer

  Kaempferide may decrease CLDN2 expression mediated by the inhibition of Akt. We have been reported that CLDN2 expression is decreased by LY-294002, a PI3K/Akt pathway inhibitor (Hichino et al., JBC 2017). Therefore, we think Akt inhibitors cannot abrogate the effects kaempferide.

Comment 7

It is needed to explain how propolis extract and kaempferide can decrease claudin-2 protein and mRNA levels.

Answer

  Following your suggestion, we described the mechanism of propolis extract and kaempferide. Please see lines 339.

Comment 8

Overall the paper is interesting although the potency of kaempferide seems marginal and in vivo test is additionally needed. The limitation of this study should be discussed carefully.

Answer

  Following your suggestion, we discussed the importance of in vivo test in the Discussion. Please see line 361.

Reviewer 3 Report

Honey bee collected resin, or propolis, contains bioactive compounds that have demonstrated antibiotic (broadly), anti-inflammatory, and anticancer activity.  Here, the authors examined some bioactive compounds found in extracts of Brazilian green propolis (EBGR) against A549 human lung adenocarcinoma cells.  Specifically, they examined the effects of kaempferide (K-ride), kaempferol (K-rol), dihydrokaempferide (DK-ride), dihydrokaempferol (DK-rol), and pinocembrin (chemical constituents of EBGR) on aspects of expression and function of claudins in A549 cells.  Claudins (CLDN) facilitate the formation of intercellular tight junctions and abnormal patterns of expression are thought to play an important role in tumor metastasis and disease progression.  The data presented in this manuscript would suggest the compounds in honey bee collected resin that were tested have activity against a biomarker of tumor progression (CLDN2) through reduction in gene expression and decreased intercellular barrier. These effects may allow more efficient of chemotherapeutics, such as doxorubicin into tumor masses as modeled in this study using A549 spheroid. The following comments/suggestions are offered on this work:

Abstract: The data may support the conclusion presented in the last sentence of the Abstract. However, I would caution that the work was done using a cell culture model, not live animal or clinical trials, so to suggest that eating foods containing K-ride would prevent malignant progression of lung adenocarcinoma is not supported but warrants further investigation.

More time could be spent in the Introduction on providing background that would prepare the reader for the techniques used or research plan, especially since many of the methods are sparse or simply refer the reader to Nasako et al. 2020 or Sonoki et al. 2017. For example, the second to last paragraph of the Introduction (Lines 58-70) could be rewritten to highlight research that draws the reader closer how identifying the mechanisms is important instead of listing general groups of compounds and generic/broad “anti-“ properties of propolis. Consider moving past results reported from Lines 216-218 here.

Materials and Methods: It is very important to specify the control condition.  Was DMSO-only used as a control to exclude solvent or carrier-effect on cell responses? Were positive controls considered for experiments since viability assays at the concentrations tested don’t show inhibition.

There is a lot of data presented (better to have too much than too little at the manuscript stage). However, EBGP is an undefined, complex mixture and only a few of the bioactive compounds commonly found in EBGP were tested in this study, specifically, K-ride, K-rol, DK-ride, DK-rol, and pinocembrin. It is ok that these compounds are the main focus as relevant literature points to a need to elucidate the mechanisms of anticancer activity of these compounds (the title of the manuscript would suggest that the focus is on the effects of K-ride). Why then include the experiments that were only EBGP as there are other compounds in EBGP that could produce the additional or synergistic effects to those observed? Added to this a general recommendation: It would be important to separate the rationale for testing the specific active compounds earlier and then bringing the reader back to the bigger picture of propolis as a natural supplement or food additive in the Discussion… that natural products exist and are worth investigating/have value in human and animal health.

Results: Throughout, the figure legends tell the reader the responses of chemical concentrations are relative to the control. In most (or all) figures, the controls are reported near 100%, as expected, and it appears that results of other concentrations that are not controls appear to be absolute and not relative levels. For example, if the control was 100% and the response of a concentration other than 0 (control) was 90% (absolute amount), wouldn’t the relative amount be -10%? Would it be easier for the reader if control was set to 0 or baseline and the findings from other amounts greater or lesser than the baseline?

The summary of the statistical model should be included in each results section not just p-values.

Line 137: There doesn’t appear to be an effect of EBGP on cell viability, even at the highest concentration, not sure what is meant by “did not show cytotoxicity until a concentration of 50ug/mL? This or similar wording was a common occurrence throughout the manuscript and could be made more clear.

Figure 2A. The staining for ZO-1 shows localization at the intercellular border. Is the magnification the same for the control and the EBGP panels? It seems the EBGP panel is greater magnification.

Figure 6. Is there a more clear/less background image of the p-Akt WB? If this is intended to show reduction, then I think it could be argued that B-actin for DK-ride is lower than control and K-ride.

Discussion: Line 312 N-acteyl-L-cysteine is found in propolis?

Lines 315-327: This paragraph is high-detailed account of molecular interactions. How can this applied to our understanding of tumor biology, even inhibition of metastasis via a K-ride driven mechanism?… Can this be rewritten to guide the reader to the main conclusion of the Abstract and final sentence of Discussion.

Lines 328-338:

Line 330: Can an explanation be included on why cell viability increased in the absence of DXR (Figure 9A)?

Line 324 What is the significance of the 2,3-double bond in the C ring of DK-ride. How might this structure or the lack thereof in K-ride increase K-ride’s activity against CLDN2?

Minor:

Are K-ride, K-rol, DK-ride, DK-rol accepted abbreviations for these compounds?

Consider more formal substitutes for “so far” (Line 13) and “so on” (Line 44 and 68).

Line 59: Resin is what bees collect from stems and leaf buds. The wording could be more clear

Line 61: Include additional citations, especially since the reference pertains to veterinary medicine.

Line 65: propolis is spelled incorrectly.

Adjust y-axis scale to include smaller increments.

Figure 6. Include A and B in panels.

Line 236-237 could be moved to the Introduction (as suggested) or the Discussion, depending on how it would be reworded.

Figure 8 confocal/fluorescent images could be higher resolution. It also appears that CLDN2 didn’t colocalize with ZO-1 under DK-ride exposure as there is considerable separation is signals on the merged image. Please explain why this may be?

Discussion: Opening sentence doesn’t offer why tumor cells have higher tolerance that allows them to live in stressed conditions. Further, consider omitting the entire first paragraph except the final two sentences, which can be place in the next paragraph.

Author Response

We thank you very much for your careful reading of our manuscript and valuable comments.

Comment 1

Abstract: The data may support the conclusion presented in the last sentence of the Abstract. However, I would caution that the work was done using a cell culture model, not live animal or clinical trials, so to suggest that eating foods containing K-ride would prevent malignant progression of lung adenocarcinoma is not supported but warrants further investigation.

Answer

  Following your suggestion, we modified the sentence. Please see line 30.

Comment 2

More time could be spent in the Introduction on providing background that would prepare the reader for the techniques used or research plan, especially since many of the methods are sparse or simply refer the reader to Nasako et al. 2020 or Sonoki et al. 2017. For example, the second to last paragraph of the Introduction (Lines 58-70) could be rewritten to highlight research that draws the reader closer how identifying the mechanisms is important instead of listing general groups of compounds and generic/broad “anti-“ properties of propolis. Consider moving past results reported from Lines 216-218 here.

Answer

  Following your suggestion, we modified the Introduction. Please see lines 59-69.

Comment 3

Materials and Methods: It is very important to specify the control condition. Was DMSO-only used as a control to exclude solvent or carrier-effect on cell responses? Were positive controls considered for experiments since viability assays at the concentrations tested don’t show inhibition.

Answer

  Following your suggestion, we described the status of control condition. Please see lines 91-93. Cisplatin (500 mM) and doxorubicin (20 mM), positive control, showed cytotoxicity about 86.4% and 61.9% under the same experimental conditions, respectively.

Comment 4

There is a lot of data presented (better to have too much than too little at the manuscript stage). However, EBGP is an undefined, complex mixture and only a few of the bioactive compounds commonly found in EBGP were tested in this study, specifically, K-ride, K-rol, DK-ride, DK-rol, and pinocembrin. It is ok that these compounds are the main focus as relevant literature points to a need to elucidate the mechanisms of anticancer activity of these compounds (the title of the manuscript would suggest that the focus is on the effects of K-ride). Why then include the experiments that were only EBGP as there are other compounds in EBGP that could produce the additional or synergistic effects to those observed? Added to this a general recommendation: It would be important to separate the rationale for testing the specific active compounds earlier and then bringing the reader back to the bigger picture of propolis as a natural supplement or food additive in the Discussion… that natural products exist and are worth investigating/have value in human and animal health.

Answer

  Following your suggestion, we modified the title. In addition, we modified the Discussion. Please see lines 320 and 359.

Comment 5

Results: Throughout, the figure legends tell the reader the responses of chemical concentrations are relative to the control. In most (or all) figures, the controls are reported near 100%, as expected, and it appears that results of other concentrations that are not controls appear to be absolute and not relative levels. For example, if the control was 100% and the response of a concentration other than 0 (control) was 90% (absolute amount), wouldn’t the relative amount be -10%? Would it be easier for the reader if control was set to 0 or baseline and the findings from other amounts greater or lesser than the baseline?

Answer

  Following your suggestion, control was set to 0. Please see new figures 3, 9, and 10.

Comment 6

The summary of the statistical model should be included in each results section not just p-values.

Answer

  Following your suggestion, we described the statistical model in each figure legend.

Comment 7

Line 137: There doesn’t appear to be an effect of EBGP on cell viability, even at the highest concentration, not sure what is meant by “did not show cytotoxicity until a concentration of 50ug/mL? This or similar wording was a common occurrence throughout the manuscript and could be made more clear.

Answer

  Following your point out, we modified the sentence. Please see lines 137-139.

Comment 8

Figure 2A. The staining for ZO-1 shows localization at the intercellular border. Is the magnification the same for the control and the EBGP panels? It seems the EBGP panel is greater magnification.

Answer

  Following your point out, we replaced the representative images. Please see new figure 2.

Comment 9

Figure 6. Is there a more clear/less background image of the p-Akt WB? If this is intended to show reduction, then I think it could be argued that B-actin for DK-ride is lower than control and K-ride.

Answer

  Following your point out, we performed additional experiments. Please see new figure 6

Comment 10

Discussion: Line 312 N-acteyl-L-cysteine is found in propolis?

Answer

  N-acteyl-L-cysteine is not a component of propolis. We modified the sentence. Please see line330.

Comment 11

Lines 315-327: This paragraph is high-detailed account of molecular interactions. How can this applied to our understanding of tumor biology, even inhibition of metastasis via a K-ride driven mechanism?… Can this be rewritten to guide the reader to the main conclusion of the Abstract and final sentence of Discussion.

Answer

  Following your suggestion, we rewrote the paragraph. Please see lines 333-344.

Comment 12

Lines 328-338:

Answer

  We do not know your point out. However, we rewrote the sentence.

Comment 13

Line 330: Can an explanation be included on why cell viability increased in the absence of DXR (Figure 9A)?

Answer

  We explained the reason. Please see line 351.

Comment 14

Line 324 What is the significance of the 2,3-double bond in the C ring of DK-ride. How might this structure or the lack thereof in K-ride increase K-ride’s activity against CLDN2?

Answer

  At present, we do not know how 2,3-double bond in the C ring is involved in the inhibition of Akt. Some Akt inhibitors such as LY-294002 and Akt inhibitor XI have 2,3-double bond. We think that further studies are needed to clarify the inhibitory mechanism of these compounds.

Minor comments

Comment 1

Are K-ride, K-rol, DK-ride, DK-rol accepted abbreviations for these compounds?

Answer

  These abbreviations are used in several articles. However, we decided to spell them full.

Comment 2

Consider more formal substitutes for “so far” (Line 13) and “so on” (Line 44 and 68).

Answer

  Following your suggestion, we modified them.

Comment 3

Line 59: Resin is what bees collect from stems and leaf buds. The wording could be more clear

Answer

  Following suggestion in major comment 2, we deleted the second paragraph.

Comment 4

Line 61: Include additional citations, especially since the reference pertains to veterinary medicine.

Answer

  Following suggestion in major comment 2, we deleted the second paragraph.

Comment 5

Line 65: propolis is spelled incorrectly.

Answer

  Thank you very much for your point out. We corrected it.

Comment 6

Adjust y-axis scale to include smaller increments.

Answer

  Following your suggestion, we corrected the y-axis.

Comment 7

Figure 6. Include A and B in panels.

Answer

  We deleted “(A)” from the legend of figure 6.

Comment 8

Line 236-237 could be moved to the Introduction (as suggested) or the Discussion, depending on how it would be reworded.

Answer

  Following your suggestion, we move the sentence to the Discussion. Please see line 335.

Comment 9

Figure 8 confocal/fluorescent images could be higher resolution. It also appears that CLDN2 didn’t colocalize with ZO-1 under DK-ride exposure as there is considerable separation is signals on the merged image. Please explain why this may be?

Answer

  As you pointed out, the red signal of CLDN2 in the tight junction was weak in the dihydrokaempferide-treated cells in the old image. We replaced the representative images. Please see new figure 8.

Comment 10

  Discussion: Opening sentence doesn’t offer why tumor cells have higher tolerance that allows them to live in stressed conditions. Further, consider omitting the entire first paragraph except the final two sentences, which can be place in the next paragraph.

Answer

  Following your suggestion, we deleted the entire first paragraph except the final two sentences.

Round 2

Reviewer 1 Report

Page 3, line 173, and Figure 3:  The authors still have not defined the abbreviation “LOX-1”, so I looked it up.  LOX-1 stands for “lectin-type oxidized LDL receptor 1”.  The authors need to add this to the manuscript.

For “Comment 8. Page 6, line 190-1:  This statement is not supported by the figure.  Answer: Following your suggestion, we corrected the sentence. Please see line 199.”  Line 199 is obviously not the correct line here.

Several English language errors have been introduced by the authors in the new version of the manuscript.

Author Response

We thank you very much for your careful reading of our manuscript and valuable comments.

Comment 1

Page 3, line 173, and Figure 3:  The authors still have not defined the abbreviation “LOX-1”, so I looked it up.  LOX-1 stands for “lectin-type oxidized LDL receptor 1”.  The authors need to add this to the manuscript.

Answer

  You suggested that LOX-1 stands for “lectin-type oxidized LDL receptor 1”. However, it is incorrect. We used a fluoresce probe, LOX-1, in order to monitor hypoxic condition. Please see lines 121-123, the legend of figure 3, and attached PDF file.

Comment 2

For “Comment 8. Page 6, line 190-1:  This statement is not supported by the figure.  Answer: Following your suggestion, we corrected the sentence. Please see line 199.”  Line 199 is obviously not the correct line here.

Answer

  I apologize for my mistake. I think you pointed out the effects of kaempferol and kaempferide on cell viability. Please see lines 198-199.

Comment 3

Several English language errors have been introduced by the authors in the new version of the manuscript.

Answer

  Thank you very much for your point out. We rechecked the English language errors.

Reviewer 2 Report

The paper is well revised.

Author Response

We thank you very much for your careful reading of our manuscript.

Reviewer 3 Report

The authors have sufficiently addressed comments and suggestions.  Importantly, the figures are improved where asked to make them easier to interpret.

There are just three remaining issues:

  1. The positive control has been described sufficiently; however, please describe the negative control.  Do you consider untreated to be the negative control?
  2. Reporting the summary of the statistical model, e.g. for one-way analysis: (Fx,y = statistic, p = 0.00).  x = between group degrees of freedom, y = within group degrees of freedom, statistic is the F-statistic from the model output.  This is a brief phrase but provides much information.  Look to journal style or writing/statistical  manuals for best reporting.
  3. Minor, Line 45:  One suggestion for "things like that" in the context of the sentence could be "and other cellular interactions".  My point was to change "so on" and "so far" to be more inclusive of the context of the sentence.  So on, so far, things like that are vague.

Thank you for thoroughly addressing comments and suggestions.
